# In Search of the Holy Grail: Stem Cell Therapy as a Novel Treatment of Heart Failure with Preserved Ejection Fraction

**DOI:** 10.3390/ijms24054903

**Published:** 2023-03-03

**Authors:** Olaf Domaszk, Aleksandra Skwarek, Małgorzata Wojciechowska

**Affiliations:** Department of Experimental and Clinical Physiology, Center for Preclinical Research, Warsaw Medical University, 02-097 Warsaw, Poland

**Keywords:** heart failure, preserved ejection fraction, cardiology, stem cells

## Abstract

Heart failure, a leading cause of hospitalizations and deaths, is a major clinical problem. In recent years, the increasing incidence of heart failure with preserved ejection fraction (HFpEF) has been observed. Despite extensive research, there is no efficient treatment for HFpEF available. However, a growing body of evidence suggests stem cell transplantation, due to its immunomodulatory effect, may decrease fibrosis and improve microcirculation and therefore, could be the first etiology-based therapy of the disease. In this review, we explain the complex pathogenesis of HFpEF, delineate the beneficial effects of stem cells in cardiovascular therapy, and summarize the current knowledge concerning cell therapy in diastolic dysfunction. Furthermore, we identify outstanding knowledge gaps that may indicate directions for future clinical studies.

## 1. Introduction

Heart failure is one of the main issues of global healthcare affecting more than 64 million patients worldwide [1]. The disease is also one of the most common causes of hospitalizations. Although significant advances in cardiovascular pharmacology have been made, heart failure mortality remains substantial [2]. This highlights the need to improve our understanding of the pathophysiology to improve the diagnosis and treatment of the disease.

Patients presenting signs and symptoms of heart failure should undergo routine laboratory tests, including natriuretic peptides and echocardiography. According to the ejection fraction, heart failure is classified into two main phenotypes—heart failure with reduced ejection fraction (HFrEF) and heart failure with preserved ejection fraction (HFpEF). According to the European Society of Cardiology and the American Heart Association/American College of Cardiology, HFrEF is diagnosed in patients with EF ≤40%, whereas HFpEF is diagnosed in patients with EF ≥50%. HFmrEF (heart failure with mildly reduced ejection fraction) is characterized by an ejection fraction between 40% and 50% [3,4].

Today, there is a marked discrepancy in the prevalence of HFrEF and HFpEF. While the incidence of HFrEF significantly decreases, there is an upward trend in the incidence of HFpEF [5]. As the risk of HFpEF increases with age, population aging can partially explain this trend. Furthermore, the disease is associated with the female sex and numerous comorbidities, such as diabetes mellitus, hypertension, obesity, atrial fibrillation, chronic coronary disease, and chronic kidney disease [6]. Thus, the treatment of concomitant diseases is regarded to be particularly relevant.

HFpEF poses one of the most significant challenges of modern medicine. Within the past three decades, there was no improvement in its prognosis. Although it usually raises fewer concerns, the mortality rate of HFpEF is comparable with that of HFrEF [5]. Recently, many efficient treatment modalities of HFrEF have appeared, but there is still an extreme paucity of therapeutic options to improve the outcomes of HFpEF. As the great majority of patients with HFpEF are hypertensive, most receive drugs, such as angiotensin-converting enzyme inhibitors (ACE-I), angiotensin-receptor blockers (ARBs), *β*-blockers, or mineralocorticoid receptor antagonists (MRAs) [7]. However, as the trials focused on these medications in HFpEF are disappointing, their use in the treatment of this disease remains debatable.

For instance, in the CHARM-preserved trial, the implementation of candesartan only reduced admissions to hospital for HF [8]. The PARAMOUNT-HF trial showed sacubitril combined with valsartan intake, an angiotensin receptor neprilysin inhibitor (ARNI)nis, was associated with more significant N-terminal pro B-type natriuretic peptide (NT-proBNP) reduction and favorable alterations in novel extracellular matrix biomarkers in comparison with valsartan alone [9,10]. However, there was no substantial reduction in the hospitalization rate [7]. Regarding MRAs, spironolactone intake is associated with beneficial hemodynamic effects. The Aldo-DHF study showed a correlation between spironolactone intake and a decline in the E/e’ ratio, which proves spironolactone results in a significant diastolic correction [11]. Importantly, none of the investigated treatments met the primary endpoints of the trials—namely a decrease in overall or cardiovascular mortality.

Due to their proven efficacy, the sodium-glucose co-transporter-2 inhibitors (SGLT2i) are already accepted for HFrEF treatment. Furthermore, new indications for its use have appeared. As insulin resistance is one of the major risk factors for HFpEF, researchers hope SLGT2i is effective in the context of the treatment of the disease.

The EMPEROR-preserved trial was the first large-cohort double-blind, placebo-controlled study investigating an SGLT2i for HFpEF, namely empagliflozin. The study achieved its primary endpoint: a composite of cardiovascular death or heart failure hospitalization, although it was related mainly to the decreased number of heart failure hospitalizations. The difference in the overall death rate and the number of hospitalizations between both groups was statistically non-significant [12].

Another SGLT2i, dapagliflozin intake, is associated with favorable outcomes in patients with HFpEF and HFmrEF. It improves the quality of life as well as exercise tolerance [13]. The drug was also tested in the DELIVER trial in which it met the primary composite outcome of unplanned hospitalizations and cardiovascular mortality. However, this effect was primarily driven by the decrease in HF hospitalizations and not mortality [14].

To summarize, to date, there are few available drugs that alleviate the symptoms of HFpEF, and none has a proven efficacy in mortality improvement. Accordingly, a significant breakthrough is still awaited. Therefore, it is essential to develop etiology-based treatment strategies to improve the prognosis.

Due to their numerous salutary traits, stem cells have attracted significant attention from researchers in recent years. Their application may be highly beneficial in a multitude of various diseases, including cardiovascular diseases, such as: coronary artery disease, dilated cardiomyopathy, and pulmonary hypertension. There is also a growing body of evidence concerning its possible efficacy in the treatment of HFpEF. The purpose of this review is to comprehensively describe the presumed beneficial effects of stem cells and to summarize the current state of knowledge regarding their use in HFpEF.

## 2. Pathophysiology of HFpEF

To better understand the role of stem cells in the treatment of HFpEF, it is crucial to know its multifactorial pathophysiological background. Among multiple components, it is assumed that immunological reactions, fibrosis, and microvascular dysfunction may play a central role in the development of the disease.

Macroscopically, HFpEF is characterized by myocardial stiffness and impaired relaxation [3,15]. The echocardiographic hallmarks of these phenomena are increased filling pressures and diastolic dysfunction. These observations were the basis for the paradigm of purely mechanistic causes of HFpEF [16]. This theory was limited to the assumption that an increased afterload exerts increased stress on the myocardium of the left ventricle, forcing the heart to generate elevated left ventricular pressure to maintain the ejection fraction, the long-term consequence of which is structural remodeling.

The common histopathological findings of HFpEF are interstitial fibrosis, hypertrophy, decreased density, inflammatory infiltrate, and impaired autoregulation of microcirculation [17,18,19,20]. Furthermore, cardiomyocyte hypertrophy, energetic abnormalities, increased oxidative stress, and endothelial dysfunction are also observed [15,20]. This clearly shows the pathophysiology of HFpEF is not limited to its mechanical background, and there are many factors whose complex interplays underlie its clinical manifestation.

Today, the inflammatory hypothesis of HFpEF is becoming more relevant. The immunological dysfunction may be a stand-alone condition. However, as the epidemiological data shows an association of HFpEF with systemic inflammation due to the previously mentioned comorbidities and aging, it may be an organ manifestation of a systemic pathology.

The pathogenesis of HFpEF has been explored only to a certain degree. The knowledge about the development of the disease is mainly derived from animal models. Nevertheless, it has partially enabled researchers to fill the existing knowledge gaps to complete the mechanistic theory. Based on the available research, we divided the pathophysiological mechanisms into three large groups, namely:(1)Dysfunction caused by microvasculature defects due to comorbidities;(2)Dysfunction caused by inflammatory activity within the heart as a response to mechanical stress;(3)Dysfunction caused by intrinsic pro-inflammatory activity due to clonal hematopoiesis.

## 3. Dysfunction Caused by Microvasculature Defects Due to Comorbidities

Several chronic diseases may increase the risk of HFpEF by inducing a hyperinflammatory state characterized by the elevated serum concentration of interleukin 6 (IL-6), tumor necrosis factor-α (TNF-α), soluble suppression of tumorigenicity 2 (sST2), and pentraxin 3. It may impair the endothelial function and increase the expression of endothelial adhesion molecules, such as the vascular cell adhesion molecule (VCAM) and the intercellular adhesion molecule (ICAM), on endothelial cells [21]. These proteins are essential for attracting circulating leukocytes.

Subsequently, the inflammation is aggravated by macrophage infiltration. Cardiac macrophages may be divided into two subsets playing distinct roles. The circulatory macrophages may be distinguished from the resident cardiac macrophages by C-C chemokine receptor type 2 (CCR2) positivity [22]. Resident cardiac (CCR2-) macrophages are distributed in the myocardium already at the embryonic stage where they regulate tissue homeostasis and the repair mechanisms [22]. On the contrary, the hematogenous (CCR2) macrophages are associated with the induction of inflammation, interstitial fibrosis, and diastolic dysfunction. They display pro-inflammatory activity in two ways:(1)Directly by interleukin 10 (IL-10) and transforming growth factor β (TGF-β) production, which stimulates fibroblast activation and collagen deposition;(2)Indirectly by antigen presentation and T cell activation.

There is a positive feedback loop between T lymphocytes and macrophages. As mentioned above, the VCAM and ICAM proteins facilitate T lymphocyte infiltration. The T cells release IFN-γ, which attracts macrophages. Macrophages thereafter present antigens to the T lymphocytes. T cells, in turn, enhance the expression of adhesion molecules on the endothelium, leading to the massive influx of immune cells into the perivascular space [22]. This could presumably be a future therapeutic target. There is one example of a successful intervention with abatacept in the murine model of heart failure, which is a medication binding the costimulatory molecules of the antigen presenting cells (APC) blocking their interaction with the T cells. By preventing T lymphocyte activation, the treatment decreased the deleterious macrophage influx in this experimental trial [23].

Pro-inflammatory cytokines can also activate nicotinamide adenine dinucleotide phosphate oxidase, increasing reactive oxygen species (ROS) production, which stimulates the proliferation and activation of collagen-producing myofibroblasts [17]. ROS, due to a reaction with nitric oxide (NO), decreases its bioavailability and stimulates the formation of peroxynitrite (ONOO−). This dysregulation induces so-called nitrosative stress, which is associated with myocardial dysfunction and microvascular endothelial inflammation, further aggravating diastolic dysfunction [24,25].

NO is also crucial for the formation of cyclic guanosine-3,5-monophosphate (cGMP), a molecule indispensable for activating cGMP-induced protein kinase (PKG). The NO-cGMP-PKG signaling not only triggers vascular smooth muscle cell relaxation but also via titin phosphorylation and may have a protective effect in HFpEF. Titin is one of the main cytoskeletal proteins and is responsible for the recoil of the sarcomere early after a contraction is completed, which translates into the diastolic distensibility of the cardiomyocytes. PKG phosphorylates the N2B segment of titin and in this way, increases its activity and compliance. If the process is interrupted, the cardiomyocyte resting tension increases, leading to diastolic dysfunction [26,27,28].

The importance of this mechanism may be emphasized by the fact that coronary infusions of NO donors could diminish the diastolic left ventricular stiffness in patients with aortic stenosis or dilated cardiomyopathy [25]. NO secretion also has multiple collateral protective effects, which are important for maintaining cardiovascular health, among which the most important are:(1)Inhibition of platelet aggregation and thrombus formation by the decrease of P-selectin expression on platelets and Willebrand factor-mediated platelet aggregation adhesion [29,30];(2)Immunomodulatory properties due to the decrease in the production of pro-inflammatory type 1 T helper cell cytokines (IFN-γ and IL-2) [31,32];(3)Prevention of the deleterious proliferation of vascular smooth muscle cells by ubiquitin-conjugating enzyme H10 degradation [33];(4)Inhibition of the cytokine-induced expression of VCAM-1 and monocyte chemoattractant protein-1 (MCP-1) [32,34].

## 4. Dysfunction Caused by Inflammatory Activity within the Heart as a Response to Mechanical Stress

The next hypothesis does not undermine the previous one, although it identifies a different leading cause underlying the whole cascade of immunological activation within the myocardium. However, the final steps, i.e., the influx of immune cells into the myocardium, remain common for both mechanisms.

As HFpEF is strongly associated with hypertension, it has long been recognized that the pathophysiology of this condition consists of the myocardial response to increased mechanical stress, i.e., pressure overload. However, the effect of elevated blood pressure on the heart is much more complex and encompasses many immunological reactions.

It has been detected that, due to mechanical stress, CCR2 ligands, such as MCP-1, MCP-2, chemokine (C-C motif) ligand 12 (CCL12), and stromal cell-derived factor 1α (SDF-1α) are overexpressed [17,22,35]. These ligands are crucial for the penetration of monocytes from the vessel lumen through the endothelium into the perivascular space. This activity of macrophages may be explained as an attempt to react promptly to possible damage to the cardiac tissues and an attempt to modify the mechanical properties of the tissue to become more robust. However, it may elicit deleterious immunological processes leading to the previously described myocardial remodeling.

The stress induced by hypertension is not limited solely to the cardiovascular system. It should be regarded rather as a systemic phenomenon. Importantly, there is a correlation between diastolic dysfunction and increased bone marrow and splenic signal in 18F-FDG PET/CET [17]. Thus, hypertension and pressure overload activate hematopoietic stem and progenitor cells, which proliferate at higher rates and are subsequently released into the circulation, further aggravating inflammation.

In response to structural damage, the heart releases alarmins, i.e., endogenous molecules that activate immune cells in response to cellular damage. They are subsequently transported into the spleen where they activate pro-inflammatory macrophage populations. The released macrophages subsequently infiltrate the myocardium, which intensifies the inflammation. In the murine model, the alarmins released by the damaged cardiomyocytes cause splenic morphological changes (e.g., an increase in the number of follicles and the size of germinal centers and marginal zone) as well as a shift in the population of immune cells within the organ over the course of heart failure, which reflects the splenic involvement in this condition. Additionally, an increase in the number of subtypes of dendritic cells in the spleen, blood, and the heart was observed. In the same study, a splenectomy reversed cardiac remodeling and decreased macrophage and dendritic cell infiltration of the myocardium [36].

This demonstrates the importance of the so-called cardiosplenic axis, which is a bidirectional route of the immune cells. It may be relevant in maintaining homeostasis; however, if dysregulated, it may become pathological and lead to maladaptive cardiac remodeling. It may also constitute a possible future therapeutic target in HFpEF.

## 5. Dysfunction Caused by Intrinsic Pro-Inflammatory Activity Due to Clonal Hematopoiesis

Clonal hematopoiesis of indeterminate potential (CHIP) is characterized by multiple mutations in hematopoietic stem cells, resulting in the clonal expansion of one subpopulation originating from a single founding cell sharing its DNA alterations [22]. The effects may be explained by the example of the TET2 gene. It is usually involved in the epigenetic regulation of the inflammation and inhibits the repression of the expression of pro-inflammatory factors, e.g., IL-6, by recruiting histone deacetylase (HDAC2). This protein leads to histone deacetylation, which in turn terminates the gene transcription [22]. It enables the resolution of the inflammation and serves as a counterbalance for the pro-inflammatory processes. Consequently, silencing the mutation of TET2 and other CHIP genes leads to the dysregulation of the inflammatory response in favor of pro-inflammatory factors.

These processes tend to accumulate with age, with CHIP mutations being detected in 1% of individuals under the age of 40, but approximately 10–20% of those over the age of 70 [37]. A retrospective study on patients with coronary heart disease and healthy controls found CHIP mutations are associated with a twofold increase in the risk of the incidence of CHD and a four-fold increase in the risk of myocardial infarction compared with patients without CHIP mutations [38]. Therefore, it is considered that CHIP may partially explain the link between aging and the increased incidence of cardiovascular diseases.

It has been hypothesized that a similar link exists between CHIP mutations and HFpEF. The silenced TET2 gene does not inhibit the pro-inflammatory activity of macrophages, which then stimulate the production of molecules, such as IL-1, IL-6, chemokines (e.g., single C-X-C motif chemokines: CXCL1, CXCL2, CXCL3), and platelet factor 4 [22]. Il-6 is thought to be pivotal in the pathogenesis of HFpEF. Its excessive secretion may be responsible for endothelial dysfunction and the previously mentioned macrophage activation. Two studies on the animal models of HFpEF have shown mice with transplanted bone marrow originating from TET2-deficient mice developed an HFpEF phenotype, including cardiac hypertrophy and fibrosis [39,40]. In this model, genetic mutations accumulating with age lead to immunological imbalance and immune cell infiltration within the myocardium, resulting in its deleterious remodeling (Figure 1).

## 6. Stem Cells as a Promising Treatment for Cardiovascular Diseases

Stem cells are undifferentiated cells that can differentiate into specialized tissue cells. Their other unique property is self-renewal, i.e., the ability to replicate and generate new stem cells. We can differentiate embryonic and adult stem cells. Embryonic cells are pluripotent, i.e., they can differentiate into any mature somatic cells. They are abundantly present only during embryonic development; hence, it is almost unfeasible to exploit them in clinical practice. Adult stem cells are present in various tissues and show promising potential. They are considered unipotent, oligopotent, or multipotent, i.e., they have the capability to differentiate into a specific cell type, a few cell types, or multiple cell types of one lineage.

In clinical studies on cardiovascular diseases, researchers mostly use endothelial progenitor cells (EPCs—which are circulating cells expressing surface cell markers that are present also on vascular endothelial cells), mesenchymal stem cells (MSCs—typically derived from bone marrow, adipose tissue, or the umbilical cord), and cardiac stem cells (CSCs—isolated from the endomyocardial biopsy specimens). Interestingly, it also became possible to obtain so-called induced pluripotent stem cells (iPSCs) directly from any type of somatic cells. All these types of stem cells exert enormous reparative and regenerative effects that could be utilized in the treatment of HFpEF.

## 7. Endothelial Progenitor Cells

EPCs originate from the bone marrow and are characterized by expressing CD133, CD34, and the vascular endothelial growth factor receptor-2 [41]. Their undoubted advantage is that they can be noninvasively supplied from peripheral venous blood. In hypoxia, there is a significant increase in the release of several factors, including hypoxia-induced factor-1(HIF-1), vascular endothelial growth factor (VEGF), and SDF-1, which are considered to play a pivotal role in EPC activation and migration [42,43]. After EPC mobilization, they can differentiate into adult endothelial cells exerting immense angiogenic and vasculogenic properties [42]. Moreover, EPCs may play another relevant role. Due to the expression of endothelial NO synthase (eNOS), they may regulate the vascular tone [44].

In clinical studies, the properties of EPCs were mostly examined in patients with coronary artery disease. This feasible and safe treatment modality seems to be highly efficient in preventing complications due to acute myocardial infarction. The EPC injection may attenuate deleterious post-infarction remodeling by enhanced neovascularization [45]. Thereafter, reduced infarct size results in significant hemodynamic improvement [46,47]. Furthermore, EPCs have demonstrated efficiency in severe cases of patients for whom all available treatment methods (i.e., percutaneous coronary intervention or coronary artery bypass grafting) were already exhausted or for those who are not suitable for those treatment modalities [48,49]. EPCs may also alleviate symptoms of chronic myocardial ischemia. Transendocardial injection of CD133 may reduce the frequency of angina episodes and improve the quality of life [50].

It was also hypothesized that EPC-capture stents, i.e., stents covered with CD34 antibodies that bind EPC from the peripheral blood, might facilitate healing by promoting neointima formation. However, the results so far of the implementation of EPC-capture stents do not support this thesis [51].

Additionally, the vasodilatory potential of EPCs may be exploited in clinical practice. In patients with pulmonary arterial hypertension, there is a significant decrease in pulmonary resistance observed after the delivery of EPCs overexpressing eNOS. This is associated with an increase in the quality of life and in exercise capacity [52].

In individuals with HFpEF, a decrease in circulating EPCs is observed. It is postulated that microvasculature alterations are one of the most relevant components inducing HFpEF [53,54]; therefore, it may be suspected that EPCs, due to vasculogenesis stimulation, improvement of endothelial dysfunction, and ability to increase NO, are presumably highly effective candidates for HFpEF treatment.

## 8. Mesenchymal Stem Cells

MSCs are usually obtained from bone marrow or fat tissue. Today, their isolation from the peripheral blood has also become possible, which appears to be a promising method [55]. They are a heterogenous population of cells, which exhibit a regenerative capacity due to their ability for multilineage mesenchymal differentiation into osteocytes, adipocytes, chondrocytes, myocytes, cardiomyocytes, fibroblasts, myofibroblasts, epithelial cells, and neurons [56,57]. Because of their complex mechanism of action, there is growing interest in MSCs in cardiology research. MSC treatment may be beneficial in cardiac diseases as they elicit antiapoptotic, immunomodulatory, proangiogenic, and antifibrotic effects [58].

MSCs, by the increased expression of B-cell lymphoma 2 gene (BCL2), survivin, and HIF-1, may have a strong antiapoptotic potential. This effect is particularly intensified in hypoxic conditions [59,60]. MSC therapy may also be associated with a decrease in cysteine-aspartic acid protease-3 expression, which plays a role in programmed cell death [60]. It is also considered that MSCs secrete exosomes enriched in pre-micro RNA (miRNA) which may have a cardioprotective effect [61,62].

Another favorable effect of MSCs is they also display immunomodulatory activity, which may be beneficial and is particularly relevant in HFpEF. Experimental studies revealed MSC therapy may diminish the inflammatory infiltrate of the myocardium. Moreover, they have the ability to modify the level of multiple cytokines, which may establish conditions facilitating myocardial regeneration [63,64].

The angiogenic capacity of MSCs associated with the release of VEGF, which is especially aggravated in hypoxia and inflammation, may result in increased vascular density [65,66]. Moreover, it was also demonstrated that MSCs can differentiate into endothelial-like cells, which indicates their potential in patients with endothelial dysfunction as an underlying mechanism of cardiac disease [67].

The following unique feature of MSCs that may be used in cardiac diseases is their antifibrotic activity. By stimulating the activity of several metalloproteinases (MMP), especially MMP-2 and MMP-9, MSCs may significantly reduce the deposition of collagen fibers [68]. Furthermore, as collagen deposition reduces the release of VEGF, the antifibrotic effect may additionally stimulate angiogenesis [69].

Thus far, the effects of MSCs on cardiovascular diseases have been extensively examined in clinical surveys, and the results have been summed up in meta-analyses. MSC treatment yields a significant hemodynamic improvement in acute myocardial infarction, especially if the transplantation is performed within the first week. MSCs are beneficial regardless of the cell delivery route, i.e., trans-endocardial injection shows a similar efficiency to intracoronary infusion [70,71].

Pooled data analysis demonstrated MSC treatment is also highly effective in patients with HFrEF. It results in a decrease in the incidence of readmission and improved exercise capacity as well as an improvement on the in New York Heart Association (NYHA) scale. Although MSCs appear to be a promising etiology-based treatment strategy for HFpEF, there are no clinical studies to date concerning their application in the treatment of the disease [72].

## 9. Cardiosphere-Derived Stem Cells

Cardiospheres are multicellular clusters generated from endomyocardial biopsy specimens. They are composed of CSCs and myofibroblasts in the center, surrounded by differentiated supporting cells, such as vascular smooth muscle and endothelial cells. The outer layer protects the stem cells against oxidative stress and is crucial for maintaining their self-renewal ability [73]. Cardiospheres are sources of cardiosphere-derived cells (CDCs), which due to their reparative properties, may inhibit the progression of cardiovascular diseases or even reverse the underlying abnormalities.

Due to the release of various microRNAs (miRNAs) and extracellular vesicles, CDCs have a strong protective effect. Among those factors, the most relevant are miRNA-146a and the pregnancy-associated plasma protein-A (PAPP-A) containing exosomes exerting potent anti-apoptotic activity [74].

Cardiac progenitor cells, similar to the two previously mentioned stem cells, may also promote angiogenesis by the release of VEGF and insulin growth factor-1 (IGF-1) [75]. An increase in VEGF combined with TGF-β downregulation may be responsible for the strong antifibrotic effect of CDC [76].

CDC treatment may also exert an immunomodulatory effect. It is considered that they can switch the pro-inflammatory M1 macrophage phenotype secreting cytokines, such as TNF-α, IL-1, and IL-6, into the anti-inflammatory M2 phenotype, ensuring favorable conditions for tissue repair. The CDCs also exert an immunosuppressive effect by decreasing neutrophil recruitment [77].

Based on experimental studies, CDC therapy may be associated with the greatest benefits in the treatment of cardiovascular diseases [78]. However, as it is a relatively new method, the data concerning its use remains restricted. To date, the regenerative potential of CDCs has been mostly investigated in patients with ischemic heart diseases [79]. The delivery of CDCs is associated with a significant reduction in scar size and an increase in viable heart mass. As a consequence, the therapy results in increased ejection fraction [80].

## 10. Induced Pluripotent Stem Cells

PSCs were first generated in 2006 by Yamanaka et al., from fibroblasts. The researchers induced pluripotency of the differentiated somatic cells by using four transcriptional factors, including Oct3/4, Sox2, Klf4, and c-Myc [81]. This innovatory discovery was followed by the rapid development of iPSC research, which made it possible to obtain iPSCs from almost any type of adult somatic cells. It became even feasible to generate the iPSCs from cells obtained in a minimally invasive manner from easily accessible cell sources, including T lymphocytes from the peripheral blood, renal tubular cells from urine samples, and keratinocytes from hair follicles [82,83,84].

By using specific protocols, iPSCs may be subsequently differentiated into any cell type, including cardiomyocytes, which potentially means a biopsy would not be necessary to generate CSCs in the future. This could be especially advantageous in conditions associated with myocyte loss (i.e., acute coronary syndrome) to reverse or repair heart damage [85]. iPSCs may also differentiate into arterial-like endothelial cells producing large amounts of NO, suggesting their possible applicability in HFpEF [86,87]. Human iPSC-derived endothelial cells have already been tested in animal models, revealing their injection may stimulate heart vascularization in vivo [88,89].

Even though iPSCs represent a very promising method of cardiac regenerative medicine, it is still in its infancy. More experimental studies are required to develop efficient and safe methods for obtaining iPSCs before their use in clinical trials.

## 11. Evidence from Experimental Studies

Due to their numerous unique properties, researchers noticed stem cells might also be utilized in the treatment of HFpEF. In experimental studies, different models of HFpEF have been used together with various kinds of stem cells (i.e., MSCs, CDCs, and EPCs) with success (summarized in the Table 1.). However, it is worth mentioning that existing experimental models of HFpEF are imperfect and do not fully reflect the complexity of the disease; therefore, these results should be interpreted with caution.

Based on the study by Kelm et al., the use of adipose-derived stromal vascular fraction (SVF), which is a source of both EPCs and MSCs, may be efficient in HFpEF. The SVF treatment significantly ameliorated diastolic dysfunction in rats of an advanced age. Additionally, it was associated with favorable alterations in coronary arteries, i.e., greater coronary flow reserve (CFR) and decreased left wall thickness [90].

In mice, in the Van Linthout et al., study, HFpEF was induced by diabetes due to the application of streptozotocin. Then, placenta-expanded MSC-like cells were injected intravenously. The study demonstrated its immunosuppressive as well as antifibrotic properties. There was a significant decrease in the cardiac expression of the pro-inflammatory factors (TGF-ß1 and IFN-γ) and an increased anti-inflammatory Treg cell count. There was also a decrease in cardiac fibrosis via enhanced titin phosphorylation and an increase in arteriole density. A limitation of this study was echocardiography was not performed. However, it was concluded the treatment resulted in diastolic improvement as an in vitro analysis showed a lower passive force of cardiomyocytes isolated from mice receiving placenta-expanded MSC-like cells [64].

HFpEF can also be obtained in rats receiving a high-salt diet due to pressure overload. In the Gallet et al., study, animals were fed a high-salt diet for 6–7 weeks. They then received CDC intracoronary or a placebo. As a result, significant hemodynamic improvements were observed, i.e., normalization of the left ventricle relaxation and left ventricle diastolic pressure in comparison with the placebo-treated counterparts. In the CDC-treated group, a significant fibrosis reduction was demonstrated due to a decrease in collagens 1 and 3 content.

Furthermore, the treatment was associated with decreased pro-inflammatory cytokines and reduced myocardial macrophage and leukocyte infiltration. CDC therapy also enhanced cardiomyocyte proliferation and due to increased microvascular density, ensured a favorable microenvironment for their survival and optimal function. These benefits resulted in a decrease in lung congestion and increased survival [63].

These findings were confirmed by the de Couto et al., preclinical trial. In this study, an analogous model was used, and due to CDC treatment, consistent results were obtained. However, the in-depth investigation revealed some additional benefits of CDC treatment. Due to an improvement in eNOS activity and a reduction in ROS, vasodilatation was restored. The therapy also reduced oxidative stress and VCAM-1 expression, resulting in decreased macrophage attachment in-vitro [91].

Rieger et al., carried out the first study concerning stem cell therapy in HFpEF in a novel large-animal model. In this randomized, placebo-controlled, blind trial, HFpEF was induced in swine by embolization-mediated nephrectomy. Then, the 26 Yorkshire pigs were subdivided into four groups. The animals recruited to three of the groups received different types of stem cells, whereas the last group was a control group. The stem cells were injected into the renal artery by an angiogram and MRI.

Promising outcomes were obtained in the porcine group receiving a combination of stem cells, i.e., MSCs and kidney-derived stem cells (KSCs). In this group, a decrease in end diastolic pressure-volume relationship (EDPVR) was detected, which reflects a reduction in distensibility and diastolic function improvement.

Additionally, a beneficial effect on chronic kidney disease was observed. There was an improvement in the glomerular filtration rate (GFR) in the group receiving both MSCs and KSCs, presumably due to the downregulation in genes associated with fibrosis, apoptosis, and inflammatory responses [92].

## 12. Clinical Studies

Promising results from a few experimental studies and the improvement of the diastolic parameters in diseases, such as chronic myocardial ischemia, HFrEF, and dilated cardiomyopathy due to cell therapy, have prompted researchers to examine the efficiency of stem cells in HFpEF [93,94,95].

The first results of the application of stem cell therapy in HFpEF were published by Frljak et al. In this groundbreaking prospective crossover research, 30 patients were treated with the standard medical treatment for HFpEF for six months and then underwent a transendocardial CD34+ cell transplantation. The CD34+ cells were injected transendocardially into the areas of diastolic dysfunction [96]. The six-month follow-up examination showed significant improvement due to the cell therapy, represented by a decrease in NT-proBNP levels and an improvement in the six-minute walk test distance. Echocardiographically, an improvement was observed in the E/e’ ratio and the local systolic strain in the myocardial segments, which were injected with stem cells, but no significant change was observed in the global longitudinal strain (GLS).

## 13. Proposed Mechanism of the Therapeutic Effect of Cell Therapy in HFpEF

Based on experimental studies, stem cell transplantation has a potent immunomodulatory effect, which may be particularly beneficial in the treatment of HFpEF. It has been detected that stem cell therapy may suppress the inflammatory state by downregulation of pro-inflammatory factors, such as TNF-α, IL-1β, IL-6, and MCP-1 [97]. This may be explained by the fact that stem cells, especially MSC, can polarize monocytes or M1 macrophages into M2, which may be an effect of the release of stem cell-derived extracellular vesicles inducing this transition [98].

The immunomodulatory effect, especially a decrease in TGF-ß and in the macrophage infiltration, may also explain the antifibrotic properties of stem cell therapy. Moreover, MSCs, because they promote MMP-2 and MMP-9 secretion, can inhibit extracellular matrix remodeling associated with collagen and fibronectin accumulation [99]. The antifibrotic effect may be relevant not only in the context of the echocardiographic features and quality of life, but it can also reduce mortality [100].

Experimental studies showed stem cell therapy may promote angiogenesis, which appears to be particularly relevant in HFpEF [101,102]. It is supposed that the proangiogenic effect is related to the paracrine effect of stem cells, i.e., the increased expression of VEGF and angiopoietin-2 mRNA [102,103]. There have also been some attempts to induce MSC to overexpress prostacyclin to enhance their vasodilatory potential [104,105].

Stem cells can also exert a cardioprotective potential via the overexpression of BCL2, survivin, and HIF-1 [59,60]. It has also been shown the therapy demonstrates an antioxidative activity due to the upregulation of several antioxidative enzymes, including heme oxygenase-1 [106].

Therefore, stem cell therapy may become the first highly efficient etiology-based therapy due to its complex mechanism of action, as summarized in Table 2, which hopefully may increase the quality of life and reduce mortality related to HFpEF.

## 14. Discussion

Experimental studies have provided promising information concerning stem cell therapy in cardiovascular diseases. It has been shown its immunomodulatory effect may improve vasculogenesis and decrease fibrosis, ameliorating diastolic dysfunction and improving prognosis. To date, only one study investigating stem cells in HFpEF has revealed substantial clinical benefits. Further follow-up studies are necessary to assess the long-term results and possible side effects. The results of the ongoing regress-HFpEF clinical trial concerning CDC in HFpEF as well as further studies are needed to prove the efficiency and provide information concerning the safety and long-term effects of stem cell therapy of HFpEF.

As mentioned above, strategies of stem cell administration are beneficial in the treatment of coronary artery disease. Nevertheless, in HFpEF as the function of the whole left ventricle is impaired, intracoronary infusion might be more efficient than transendocardial injection. Furthermore, this significantly less invasive delivery method may presumably diminish the risk of arrhythmogenic complications by preventing local tissue injury [107,108].

We also suggest future studies in this field should focus on performing allogeneic stem cell transplantations, which may give better therapeutic effects than autologous transplantations. It is worth noting that patients with HFpEF are very frequently elderly patients, which limits the regeneration potential of stem cells in the case of their autologous origin. It is also worth noting that allogeneic stem cell transplantation may also make it possible to avoid CHIP-related mutations.

Due to insufficient data, many aspects of stem cell therapy of HFpEF still remain obscure. For instance, there is insufficient information to establish a specific treatment schedule. In cardiovascular diseases, mostly single injections of stem cells were examined. However, sequential treatment may possibly result in better outcomes.

Similarly, there is no consensus on which kind of stem cells should be administered in HFpEF. We propose tailored therapy may be considered in subsequent studies. For instance, patients with fibrosis and inflammation within the myocardium may achieve significant benefits from MSC or CDC treatment, whereas patients with a decrease in microvascular density and endothelial dysfunction are more likely to benefit from EPC treatment. Presumably, combined therapy composed of different kinds of stem cells, due to their synergistic properties, may be the gold standard of HFpEF treatment in the future. Hopefully, subsequent studies will provide comprehensive information to advance this treatment to obtain the most beneficial breakthrough stem cell therapy for HFpEF.

## 15. Conclusions

Today, there is a growing tendency in the incidence of HFpEF. However, there is no therapeutic option to modify the course of the disease, and it is still characterized by a high mortality rate. Hypotheses explaining the complex pathogenesis of HFpEF associate the development of the disease mostly with inflammation. Importantly, stem cells exert immunomodulatory properties, which as a consequence, may improve the diastolic function. There are promising results of a few experimental studies of stem cell therapy in HFpEF. However, further clinical studies are crucial to determine stem cell therapy effectiveness and safety and to develop specific clinical treatment guidelines.

## Figures and Tables

**Figure 1 ijms-24-04903-f001:**
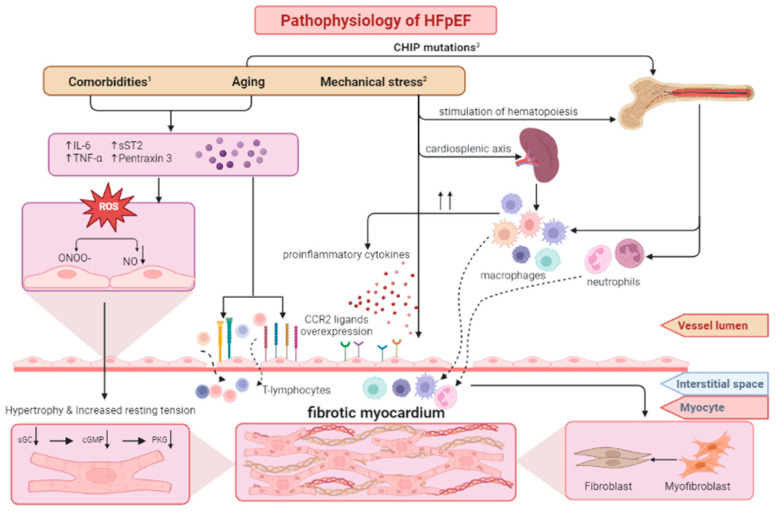
The pathomechanisms of HFpEF associated with 1—comorbidities and aging, 2—mechanical stress, and 3—CHIP mutations, leading to myocardial fibrosis and diastolic dysfunction over the course of HFpEF. Based on [21]. CCR2—C-C chemokine receptor type 2, cGMP—cyclic guanosine-3,5-monophosphate, CHIP—clonal hematopoiesis of indeterminate potential, HFpEF—heart failure with preserved ejection fraction, IL-6—interleukin 6, NO—nitric oxide, ONOO—peroxynitrite, PKG—cGMP-induced protein kinase, sGC—soluble guanylate cyclase, sST2—soluble suppression of tumorigenicity 2, TNF-α—tumor necrosis factor-α.

**Table 1 ijms-24-04903-t001:** Effects of stem cell therapy in controlled experimental studies.

Study	Model	Type of Stem Cells	Route of Delivery	Observed Effects
Kelm et al. [90]	rats of advanced age	SVF	intravenous	reduced diastolic dysfunction increased CFR decreased left wall thickness
Van Linthout et al. [64]	streptozotocin-induced diabetic mice	placenta MSC-like cells	intravenous	decreased passive force of cardiomyocytes immunosuppressive effect (decrease in TGF-ß1 and IFN-γ, increase in Treg) antifibrotic effect angiogenic effect
Gallet et al. [63]	rats fed a high-salt diet	CDCs	intracoronary	increased survival reduced diastolic dysfunction decreased hypertrophy immunosuppressive effect (decreased macrophage and leukocyte infiltration) antifibrotic effect
de Couto et al. [91]	rats fed a high-salt diet	CDCs	intracoronary	reduced diastolic dysfunction reduced macrophage infiltration improved vasodilatation reduced oxidative stress
Rieger et al. [92]	nephrectomy-induced CKD	MSCs + KSCs	via intrarenal artery	reduced diastolic dysfunction increased GFR

**Table 2 ijms-24-04903-t002:** Proposed mechanisms of stem cell therapy in HFpEF.

Effect of Stem Cell Therapy	Proposed Mechanisms
Immunomodulatory	Down-regulation of pro-inflammatory factors (TNF-α, IL-1β, IL-6, and MCP-1) Polarization of the M1 macrophages into M2 by the release of extracellular vesicles Decreased leukocyte and macrophage infiltration
Antifibrotic	Decreased TGF-ß Decreased macrophage infiltration MMP-2 and MMP-9 secretion
Proangiogenic	Increased VEGF and angiopoietin-2
Antiapoptotic	Increased survival Reduced diastolic dysfunction Decreased hypertrophy Immunosuppressive effect (decreased macrophage and leukocyte infiltration) Antifibrotic effect

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
