# Peer review of "In Search of the Holy Grail: Stem Cell Therapy as a Novel Treatment of Heart Failure with Preserved Ejection Fraction"

_ijms, 2023, doi:10.3390/ijms24054903_

Round 1

Reviewer 1 Report

1. Some of the important areas of stem cells are missing in the current manuscript. The focus is on the ejection fraction but literature that reported improved ejection fraction is fully reviewed. The references should be added and must be up to date. 

2. The MSCs section is properly described, more literature and studies should be added. 

3. IPSCs are completely ignored. 

4. The mechanism should be discussed, and a separate table must be added where references to the proposed mechanism involved in cardiac regeneration and improvement in cardiac function are cited.  

Author Response

Dear Sir or Madame, 

thank you very much for the valuable advice. It allowed us to expand the paper. We tried to improve the article as suggested-i.e.,

1.We added some details concerning stem cells, however, we did not want the paper to be very detailed as it is rather to interest the clinicians and it could presumably be a background for future studies. We also completed the references.

2. We added some literature to the MSCs section.

3. We added additional information concerning iPSCs (lines 386-405).   4. We completed the information about the proposed mechanism and generated a separate table (lines 476-502, table 2.)  

Reviewer 2 Report

In this manuscript, the authors explained the complex pathogenesis of HFpEF, delineate the beneficial effects of stem cells in cardiovascular therapy, and summarize the current knowledge concerning cell therapy in diastolic dysfunction. They also identified the remaining knowledge gaps that may propose further directions for future clinical studies.

1)    The overall writing has some formatting issues, like wording and spacing. I suggest the authors check the grammar and avoid any typos.

2)    For each section, I would suggest the authors to explain more details. The current version is a little simplified.

3)    I would suggest the authors discuss using the state-of-art single-cell technology (e.g., PMID: 35910046) and spatial transcriptomics (PMID: 36545790) that have been used to investigate the role of different types of cells, which helps expand the scope of the review.

Author Response

Dear Sir or Madame,  thank you very much for the valuable advice. It allowed me to expand my paper. I tried to improve my article as suggested-i.e., 1. We checked the grammar and improved formatting. Linguistic corrections were made by a qualified translator 2. More details were given in different sections, like in lines 386-405, lines 476-502, table 2 3. Using state-of-art single-cell technology and spatial transcriptomic have not been discussed as in our opinion, this topic goes beyond the scope of the paper and it is too detailed for the majority of the expected audience, which we believe, will be cardiology scientists.

Round 2

Reviewer 1 Report

The authors successfully addressed the review comments raised during the first round of review. Therefore, the Manuscript is accepted in its current form. 

Reviewer 2 Report

I have no more concerns about this manuscript.